# Exercise Modifies the Brain Metabolic Response to Chronic Cocaine Exposure Inhibiting the Stria Terminalis

**DOI:** 10.3390/brainsci13121705

**Published:** 2023-12-11

**Authors:** Colin Hanna, Rutao Yao, Munawwar Sajjad, Mark Gold, Kenneth Blum, Panayotis K. Thanos

**Affiliations:** 1Behavioral Neuropharmacology and Neuroimaging Laboratory on Addictions, Clinical Research Institute on Addictions, Department of Pharmacology and Toxicology, Jacob School of Medicine and Biosciences, State University of New York at Buffalo, Buffalo, NY 14203, USA; 2Department of Nuclear Medicine, State University of New York at Buffalo, Buffalo, NY 14203, USA; 3Department of Psychiatry, Washington University School of Medicine, St. Louis, MO 63110, USA; 4Division of Addiction Research and Education, Center for Sports, Exercise and Global Mental Health, Western University Health Sciences, Pomona, CA 91766, USA; 5Department of Psychology, State University of New York at Buffalo, Buffalo, NY 14203, USA

**Keywords:** rat, 18F-FDG fluorodeoxyglucose, positron emission tomography, aerobic exercise, glucose metabolism, statistical parametric mapping, cocaine

## Abstract

It is well known that exercise promotes health and wellness, both mentally and physiologically. It has been shown to play a protective role in many diseases, including cardiovascular, neurological, and psychiatric diseases. The present study examined the effects of aerobic exercise on brain glucose metabolic activity in response to chronic cocaine exposure in female Lewis rats. Rats were divided into exercise and sedentary groups. Exercised rats underwent treadmill running for six weeks and were compared to the sedentary rats. Using positron emission tomography (PET) and [18F]-Fluorodeoxyglucose (FDG), metabolic changes in distinct brain regions were observed when comparing cocaine-exposed exercised rats to cocaine-exposed sedentary rats. This included activation of the secondary visual cortex and inhibition in the cerebellum, stria terminalis, thalamus, caudate putamen, and primary somatosensory cortex. The functional network of this brain circuit is involved in sensory processing, fear and stress responses, reward/addiction, and movement. These results show that chronic exercise can alter the brain metabolic response to cocaine treatment in regions associated with emotion, behavior, and the brain reward cascade. This supports previous findings of the potential for aerobic exercise to alter the brain’s response to drugs of abuse, providing targets for future investigation. These results can provide insights into the fields of exercise neuroscience, psychiatry, and addiction research.

## 1. Introduction

Cocaine is regularly used by over 2 million people in the United States [1]. This poses many health risks, including a greater propensity for stroke, cardiovascular disease, and psychiatric/behavioral problems, such as violence, paranoia, and psychosis [2,3,4,5]. Cocaine is a highly addictive substance, and barriers to effective treatments include a high dropout rate among standard treatment programs [1]. Additionally, substance use during adolescence, including cocaine, increases the risk of developing substance use disorders in adulthood [6]. Cocaine use in humans can also induce brain changes. Adolescent substance use is also associated with negative brain changes, both structurally and functionally [6]. These structural and functional changes are present in adult substance abuse models as well [7,8,9]. Preclinical models have observed anatomical brain changes after chronic cocaine exposure. Magnetic resonance imaging found changes in the gray matter volume in male rhesus monkeys, with lower densities in the thalamus, the amygdala–hippocampal–entorhinal cortex, the parietal cortex, the insular cortex, and the orbitofrontal cortex [10]. In rodents, chronic cocaine gradually decreased metabolism in mesolimbic regions [11,12].

Cravings among cocaine users were associated with metabolic activity in the amygdala, striatum, and insula [10]. Multiple studies confirm decreased frontal metabolism in cocaine users [7,8,11]. Decreased frontal metabolism in cocaine users was found to correlate with decreased dopamine D2 receptor availability, which is known to increase cravings [7,13]. Cocaine dependence is also associated with dopamine depletion [11,14,15] and acts on the striatum via dopamine more than any other part of the brain [16,17].

Scientists are currently investigating new approaches to addiction rehabilitation utilizing both pharmacological and non-pharmacological methods [1,18,19,20]. Physical exercise is currently being observed as a potential therapeutic for drug addiction [20,21]. Exercise is a known promoter of health and wellness, improving cardiovascular health and psychiatric health, and can be used to combat obesity [22]. Exercise has been shown to modulate the brain reward system through a number of different mechanisms, including dopamine BDNF, endocannabinoids, and hormones [23]. Exercise has been shown to decrease cocaine-seeking behavior in rats as it is believed to act on similar dopaminergic circuits in the midbrain [19,21,24,25]. Additionally, exercise can have an attenuating effect on ethanol withdrawal in both sexes of rats [26]. A study by Robison et al. found that aerobic exercise can modulate dopaminergic receptor levels in the striatum. The results of this study found significant decreases in DRD1 receptors in the olfactory tubercle and nucleus accumbens shell while finding significant increases in DRD2 receptors in the caudate putamen [24]. DRD2 receptor availability [27,28,29] and sensitivity [30] are associated with reward-seeking behavior [31]. Mice that were selectively bred for wheel running showed higher basal dopamine levels and increased levels of dopamine in the nucleus accumbens after wheel running [32]. Multiple studies have confirmed the ability of physical exercise to increase levels of dopamine in the brain’s reward system [24,32,33,34,35]. Additionally, lower levels of dopamine in the reward system are associated with depressive symptoms and anhedonia [36,37,38]. In addition to direct dopamine modulation, both forced and voluntary physical exercise can increase tyrosine hydroxylase throughout the brain’s reward neuropathways [32,33,34,35,38].

Exercise has been observed influencing relapse in rats when comparing relapse results to exercise occurring in both early and late phases of abstinence [39]. It was observed that physical exercise that occurred in the early days of abstinence greatly decreased cocaine seeking, while exercise during late abstinence was ineffective at attenuating cocaine-seeking behavior [39]

The benefit of using exercise as a means of treating substance addiction is that it is a very cost-effective option. Additionally, studies have shown that exercise can be a preventative means of combating addiction, indicating that pre-exposure to exercise might decrease addiction risk [40,41,42]. In a study observing the relationship between participation in sports and health habits among high school students, the results showed that males who participated in sports were less likely to report cigarette smoking and illicit drug use (including cocaine) when compared to controls [43]. This effect on drug use was not observed in females who participated in sports [43]

Exercise has been used to aid addiction treatment in humans. In a study involving 45 subjects undergoing a 4-week inpatient rehabilitation program for substance addiction, subjects underwent a physical exercise program that included aerobics, bodybuilding, and circuit training [44]. The bodybuilding program resulted in significant decreases in reported symptoms of depression [44]. Additionally, Quigong, a form of movement-based meditation, decreased cocaine cravings and depressive symptoms in 101 cocaine-dependent subjects [45]. Lastly, in 24 individuals being treated for concurrent cocaine and tobacco addiction, researchers observed the effects of running and walking on subject physiology and abstinence [46]. Subjects either walked or ran for 30 min, three times a week for four weeks. Exercise significantly reduced resting heart rate after four weeks. There was an observed improvement from cocaine abstinence, yet the researchers reported that this improvement was not statistically significant [46]. Exercise itself is an effective psychiatric treatment when prescribed for depression and anxiety [47]. While exercise’s efficacy is equal to other interventions, it is highly dependent on patient adherence to the regimen. Unfortunately, patients are more adherent, for example, to antidepressants than running, limiting the latter’s efficacy [48]. Making exercise more reinforcing [49] may be a solution while we wait to capture the essence of exercise in a pharmaceutical or neuromodulatory intervention.

18F-FDG is the most commonly used radiotracer in PET studies [50]. This tracer can be utilized to trace regional brain changes in metabolism, which can, in turn, be related to behavioral, neuropsychological, and psychological data [50]. In a study looking at brain amygdala metabolic activity in obese women, chronic physical exercise was found to reduce stress-related amygdala activity [51]. Additionally, an FDG PET study found that exercise can inhibit the limbic system in humans by decreasing BGluM in the cingulate gyrus and the substantia nigra [52,53].

Previously, FDG PET showed how exercise can metabolically activate the caudate putamen, hippocampal subregions, and sensory cortical areas in rats [54]. In a subsequent experiment, acute cocaine was found to activate the substantia nigra and inhibit the ventral endopiriform nucleus in exercised rats [55]. This is in agreement with many previous findings that state that exercise acts similarly on dopaminergic areas of the midbrain and mesolimbic areas [33] and that activation occurs in the somatosensory cortical areas after treadmill exercise [56]. A study looking at BGluM in runners revealed increased metabolic uptake in the temporoparietal association cortex, premotor cortex, cerebellar vermis, and occipital cortex, with metabolic increases in the leg motor, thorax, and arm areas of the primary somatosensory cortex [57]. This is generally in agreement with our previous findings on exercise and FDG PET [55].

However, there have also been studies that contradict our previous findings. In rats, forced swimming was found to metabolically inactivate many regions, including the hippocampus, insula, and inferior colliculus [58], but this was not in response to chronic swimming. There have been human studies that show that aerobic exercise can inhibit metabolic uptake in the substantia nigra in older adults with mild cognitive impairment [52].

FDG PET brain imaging has also been utilized to examine the effects of cocaine in humans and animals. One study by Henry et al. looked at the effects of cocaine self-administration in rhesus monkeys after an acute dose in a cocaine-naïve state, after 60 sessions of intravenous cocaine self-administration, and after a 4-week withdrawal period [59]. A single dose of cocaine in the naïve state only induced metabolic increases in the medial prefrontal cortex, and this pattern extended through the early stages of the self-administration paradigm. As self-administration continued, BGluM activation was observed in the orbitofrontal and medial cortices, the anterior cingulate cortex, and small portions of the striatum, including the nucleus accumbens [59]. The withdrawal phase suppressed these metabolic changes, and only frontal metabolic activation remained [59]. A rodent study observed similar findings, with BGluM increases in the Cpu and prefrontal cortex after cocaine exposure [60].

In humans, 49 polysubstance users in residential treatment underwent FDG PET scans [61]. These individuals were assessed for cocaine, heroin, alcohol, MDMA, and cannabis use frequency. This study observed noteworthy inverse associations between the intensity of drug consumption for heroin, alcohol, MDMA, and cannabis and cerebral metabolism in the dorsolateral prefrontal cortex and temporal cortex. Furthermore, alcohol consumption exhibited a connection with reduced metabolic activity in the frontal premotor cortex and putamen, while stimulant usage was linked to metabolic alterations in the parietal cortex [61].

In a study by Volkow et al., 21 neurologically intact cocaine abusers underwent FDG PET scans after 1–6 weeks of cocaine abstinence [8]. Compared to controls, global rates of BGluM uptake did not differ. However, this study unveiled a trend of decreased frontal metabolism that persisted after 3–4 months of detoxification.

For the present experiment, female rats were studied, as previous work has shown that aerobic exercise was found to attenuate cocaine self-administration [62]. Differences in conditioned place preference tests were observed in a sex-specific manner [20,63,64,65,66,67]. This vulnerability might be mediated by estradiol [68]. Estradiol has been found to enhance the rewarding effects of stimulants [69,70,71]. Exercise has been found to decrease levels of estradiol in women [72,73], which suggests its protective properties against the reinforcing effects of addictive substances. Previous experiments from our lab observed the effects of physical exercise and acute cocaine exposure on BGluM in female rats [54,55]. The current experiment utilized the same exercise regimen and brain imaging protocol to assess the effects of chronic cocaine exposure between exercised and sedentary female rats.

## 2. Materials and Methods

Animals: Young adult (8 weeks) female Lewis rats (n = 16) were received from Taconic (Hudson, NY, USA). As per the standard housing protocol, rats were individually housed at ~22 °C on a 12-h reverse light/dark cycle. The dark cycle was from 6 am to 6 pm. Unlimited access to food and water was provided in the rats’ home cages. Daily handling occurred, and environmental habituation occurred for one week. This study complied with the National Academy of Sciences Guide for the Care and Use of Laboratory Animals (1996) and was approved by the University at Buffalo Institutional Animal Care and Use Committee (PROTO202100079, 5/13/22).

Exercise Regimen: A customized treadmill divided into individual plexiglass running lanes was used for forced running. The exercise regimen started at 10 min a day at 10 m/min, increasing by 10 min each day until the maximum time of 1 h was reached. The animals were given a ten-minute break after 30 min of running. This exercise regimen was maintained for 5 days per week for 6 weeks. At the conclusion of the exercise regimen, the total distance run was ~16.5 km. Sedentary rats remained in their home cages for the duration of the exercise regimen, performed as previously described [19,24,25,54].

Cocaine Treatment: Cocaine was obtained from Sigma Aldrich in St. Louis, MO, USA. The cocaine was dissolved in 0.9% saline and injected via the intraperitoneal route at 25 mg/kg. Cocaine administration occurred for 8 days (alternated with saline).

PET imaging: PET scans occurred ~2 weeks after chronic cocaine exposure. Food was restricted for 8 h to normalize blood glucose levels. Rats were then given 500 ± 115 μCi of 18F-FDG injected through the intraperitoneal route. A 30-min uptake period followed the injections, and the animals were anesthetized immediately after. Rats were anesthetized at 3% isoflurane, maintained at 1% throughout the scan. Activity was recorded using a PET R4 tomograph (Concorde CTI Siemens, transaxial resolution: 2.0 mm full width at half maximum, transaxial field view: 11.5 cm). Anesthetized rats were secured on the scanner bed for 30 min as per standard imaging protocol.

Statistical analysis: PET image analysis was conducted as previously described [55,74,75]. Briefly, scans were reconstructed using the MAP algorithm (15 iterations, 0.01 smoothing, 256 × 256 × 256 resolution) [76]. Reconstructed scans were manually coregistered onto the Schweihardt MRI template (63 slices, Paxinos and Watson stereotaxic coordinates) in the bioinformatics imaging software pMOD (Version 2.85, PMOD technologies, Fallanden, Switzerland).

Automatic coregistration and spatial normalization were performed in MATLAB software (MATLAB, Version R2018b). A statistical parametric mapping ANOVA (Voxel Threshold, K > 50) was then used to find significant differences in cluster size between the exercise and sedentary groups. Significant clusters were again fitted onto the rat brain MRI template using PMOD software (Version 4.006. These clusters were then mapped and labeled using “*The Rat Brain in Stereotaxic Coordinates*” atlas [77].

A timeline of this experiment can be seen in Figure 1, illustrating the chronic treatment of cocaine and the timing of PET imaging.

## 3. Results

A one-way ANOVA (*p* < 0.001, df = 14, K > 50) revealed the significant effects of chronic cocaine exposure in exercised rats compared to sedentary rats. Exercise and chronic cocaine only activated (BGluM increases) the secondary visual cortex, lateral area (V2L). Complete details about cluster size and statistical significance can be seen in Table 1. The cluster image of significant BGluM activation is shown in Figure 2. Exercise and chronic cocaine treatment also resulted in significant inhibition (BGluM decreases) in the paraflocculus (PFL), the eighth cerebellar lobule (8cb), the paramedian lobule (PM), the copula of the pyramis (COP), the stria terminalis (st), the stria medullaris of the thalamus, the medial and posteromedial parts of the bed nucleus of the stria terminalis (stmpm), the ventrolateral thalamic nucleus, (VL), and the primary somatosensory cortex, hindlimb region (S1HL). Complete details about cluster size and statistical significance can be seen in Table 2. The image in Figure 2 and Figure 3 shows significant clusters.

## 4. Discussion

Exercised rats only showed V2L activation after cocaine treatment. Although there is some evidence that cocaine can modulate visual processing [78], we assume this to be a result of metabolic transience attributable mostly to exercise [50]. A few previous publications have documented an increase in BGluM in sensory cortical areas after physical exercise [50,54,55,57]. This sensory cortical modulation has been observed by our group in the presence of exercise alone and after a single dose of cocaine [54,55]. Treadmill running is a task that requires visual and spatial attention. Therefore, the activation of the V2L is unsurprising.

Chronic cocaine and chronic exercise inhibited activity in various parts of the cerebellum related to eye movement and visuomotor coordination [79,80,81,82]. There are few studies that have confirmed a response to cocaine in these areas, save for a study that showed an expression of cocaine- and amphetamine-regulated transcript peptides in the PFL [83]. This inhibition can most likely be attributed to exercise. The PM is an important part of the limb motor circuit known to respond to exercise metabolically with synaptic activity and blood vessel formation [84].

The primary somatosensory cortex (S1HL) integrates information from mechanoreceptors, chemoreceptors, thermoreceptors, and nociceptors from the peripheral nervous system to map the surrounding environment [85]. Biocytin injection studies have identified pathways from the S1 to the granular and dysgranular parietal insular cortices and in the amygdaloid nuclei [86]. It receives thalamic input [87] and is involved in tactile representations, sensations, and touch (including through whiskers in mice) [88]. A study by Holshneider et al. found that treadmill walking increased blood flow to the S1HL, COP, and other associated areas [89].

Importantly, we see that exercise and cocaine inhibited parts of the brain reward cascade [13], including portions of the CPu, st, and the thalamus. The st serves as an output for the pathway from the amygdala to regions such as the hypothalamus, fornix, and thalamus [90]. St activity can be associated with an anxious temperament and increased cortisol (the stress hormone) in humans and in non-human primates [91]. The st also shares connections with the ventral tegmental area [92]. In a study by Sartor and Aston-Jones, the ventral bed nucleus of the stria terminalus (vBNST) was disconnected from the VTA via baclofen plus muscimol. The blocking of this pathway significantly reduced the preference for cocaine [92]. Considering this and previous research establishing physical exercise as a means of decreasing drug seeking [23], this inhibition may be a sign of the protective factors provided by exercise.

Lesioning of the BNST has many behavioral implications. First, it has been shown as modulating coping behavior in the presence of stress [93]. Rats with lesions in the BNST showed significantly decreased escape behavior in the forced swim test [93]. Lesions in this area are also known to attenuate the conditioned stress response in rats [94].

A subregion of the st is the stpm, which is known to be involved in stress-triggered drug relapse [95]. Inhibition of the bed nucleus of the stria terminalis via serotonin signaling was found to decrease the anxiogenic effects of cocaine [96]. A pathway between the amygdala and the st containing corticotropin-releasing factor was also found to play a mediating role in stress-induced cocaine-seeking behavior [97]. The CPu, a region in the basal ganglia/striatum, is involved in movement and goal-directed/habitual reward-seeking behaviors [98].

The VL is involved in the spinocerebellar motor pathway, and it shares connections with both the mesolimbic and motor systems [99]. An increase in cerebral blood flow was found in this area after exercise [89]. The sm of the thalamus sends projections from the forebrain to the habenula, a part of the limbic system that is highly responsive to cocaine [100]. Lastly, thalamic nuclei have been identified as moderators for striatal glutamate levels [101]. Specifically, the ventromedial motor nucleus of the thalamus (though not presently inhibited) was found to decrease cocaine-induced striatal glutamate levels when damaged [101], establishing the thalamic nuclei as key moderators of the effects of cocaine and their interaction with exercise.

We hypothesize that the exercise-induced inhibition of the st after chronic cocaine use is of high significance to addiction research. Previously, our group identified BGluM changes in brain regions associated with reward in the basal ganglia and striatum [55]. Although important, the dopaminergic pathways in these regions share functionality with motor/movement initiation and inhibition [102]. Inhibition of the st is perhaps the only region detected by statistical parametric mapping that can be considered an exclusive member of the mesolimbic/mesocorticolimbic reward pathway, in which the amygdala shares connections with the ventral tegmental area, the nucleus accumbens, and the hippocampus [103]. This circuit is important for encoding emotional value to a rewarding stimulus and shares dense connections with striatal and basal ganglia motor circuits, influencing the dopamine neuron populations of this region [49,104,105,106,107]. A paper by Torrisi reported a circuitry of the BNST that included many of our regions of interest from previous [14,15] and present experiments, including the substantia nigra, thalamus, hippocampus, and striatum [108]. This circuitry, centered around the BNST, can be reviewed in Figure 4 [108]. A proposed circuitry based on these results can be viewed in Figure 4.

Exercise has been found to decrease whole-brain BGluM uptake in humans. After 25 min of exercise, PET scanning results showed a global decrease in all measured cortical regions in 14 men subjected to bicycle exercise. The rate of metabolic uptake was negatively correlated with exercise intensity, which meant that the highest intensity exercise resulted in the lowest global metabolic uptake [109].

A proposed circuitry of our significant result clusters, based on previous findings on neuropathways and anatomical connections, can be seen in Figure 5. Functional connectivity and FDG PET imaging have been previously reported for formulating a hypothesis [55,75].

## 5. Conclusions

This paper offers insight into the brain structures and circuits involved in exercise’s effects on functional connectivity and how this connectivity is impacted in response to chronic cocaine in rats. BGluM inhibition of regions involved in the brain reward cascade [110], including regions of the thalamus, CPu, and St, were observed. Additionally, many areas that displayed changes in BGluM share dense dopaminergic connections with brain regions known to be involved in reward and addiction, such as the lateral habenula and the ventral tegmental area. This research provides targets for further mechanistic brain investigation to see if artificial inhibition of these areas results in a decrease in drug preference. Future studies can further investigate these changes to confirm if induced inhibition in these areas can help with addictive psychiatric conditions. Future studies will also observe varying intensities and types of exercise.

## Figures and Tables

**Figure 1 brainsci-13-01705-f001:**
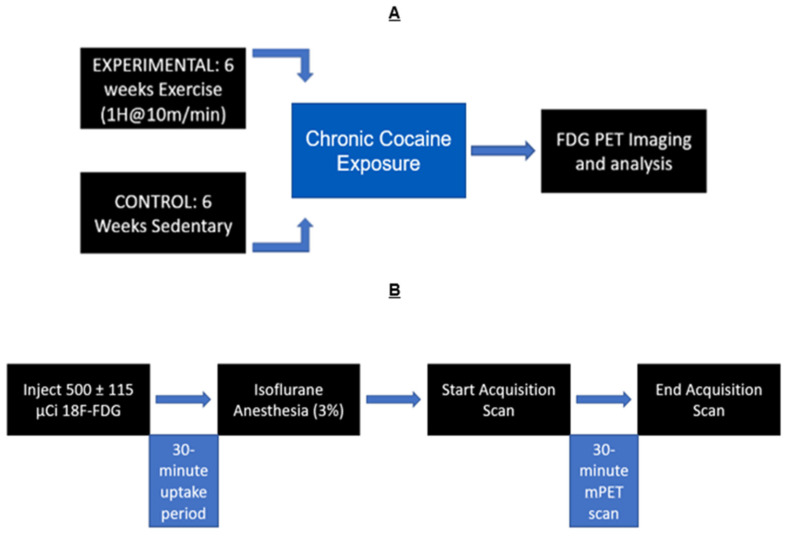
Experimental timeline: (**A**) Exercised animals ran for 6 weeks, while sedentary animals received no exercise. All animals received chronic cocaine exposure and microPET scans after the 6 weeks. (**B**) Timeline of PET scans: Rats were given [18F]-Fluorodeoxyglucose (FDG). Then, a 30 min uptake period followed. Animals were then anesthetized with isoflurane (3%) maintained throughout the duration of the 30 min PET scan (1%).

**Figure 2 brainsci-13-01705-f002:**
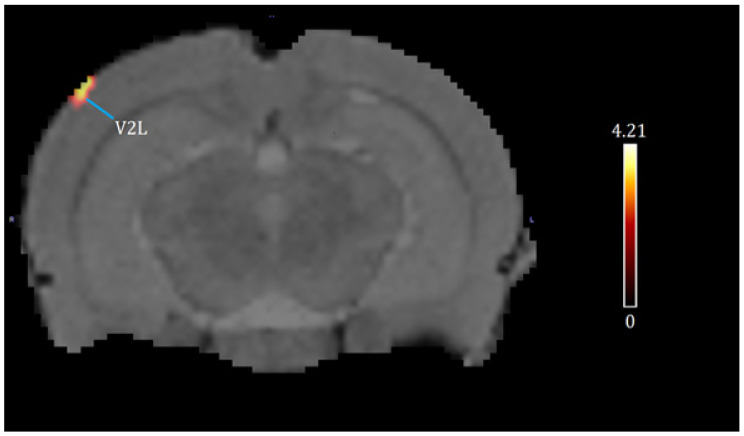
Significant activation clusters: coronal brain images with significant (*p* < 0.001, df = 14, and K > 50) metabolic increases from exercised rats compared to sedentary rats. T-values represent peak activation (t = 4.21). Red clusters indicate BGluM activation in the V2L.

**Figure 3 brainsci-13-01705-f003:**
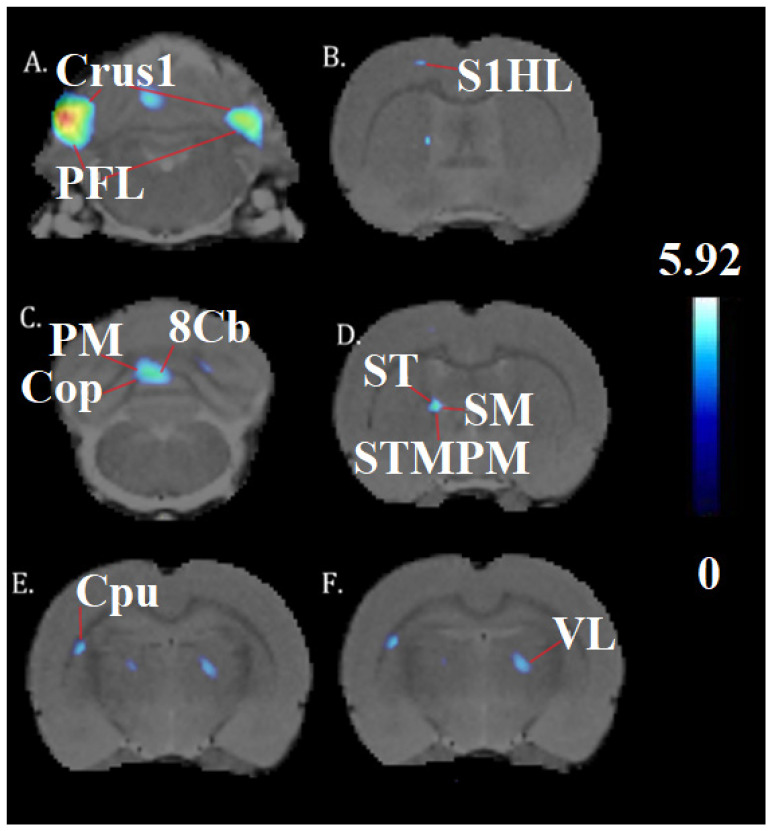
Significant inhibition clusters: coronal brain images with significant (*p* < 0.001, df = 14, and K > 50) metabolic decreases (**A**–**F**) in exercised rats compared to sedentary rats. T-values represent peak inhibition (t = 5.92). Red lines are used to label and differentiate separate anatomy. Blue clusters illustrate BGluM inhibition in the (**A**): Crus1, PFL; (**B**) S1HL (**C**) Cop, PM, 8Cb; (**D**) st, sm, STMPM; (**E**) Cpu; (**F**) VL.

**Figure 4 brainsci-13-01705-f004:**
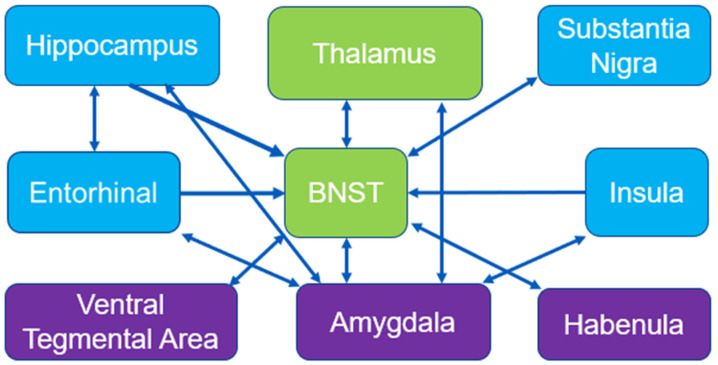
Mesolimbic circuitry and connections: BNST circuitry adapted from Torrisi et al., 2015 [108]. Blue boxes represent activated regions of interest in exercised rats compared to sedentary from Experiment 2 (acute cocaine). Green boxes represent inhibited regions of interest in exercised rats compared to sedentary from Experiment 3 (chronic cocaine). Purple boxes represent miscellaneous regions of interest relevant to reward circuitry.

**Figure 5 brainsci-13-01705-f005:**
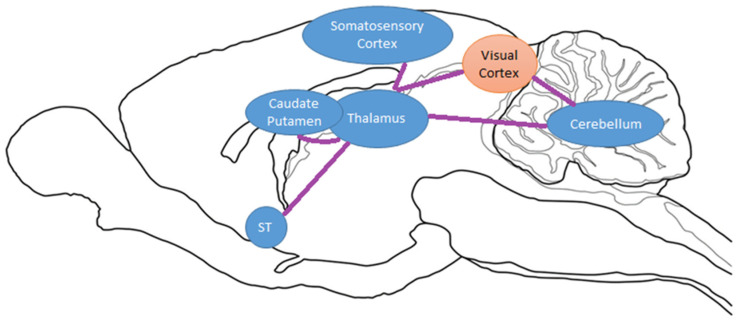
Sagittal drawing of hypothesized brain circuitry due to exercise and chronic cocaine exposure. Activated/increased BGluM clusters are shown in orange. Inhibition of BGluM is shown in blue.

**Table 1 brainsci-13-01705-t001:** Brain regions displaying significant increases (*p* < 0.001, K > 50) in BGluM after chronic exercise and chronic cocaine exposure. Cluster location is indicated (medial–lateral, anterior–posterior, and dorsal–ventral). KE represents the cluster size or number of voxels in the cluster. Each box under “Brain Region(s)” indicates a separate cluster.

Brain Region	Cluster Location (General)	Medial–Lateral (mm)	Dorsal–Ventral (mm)	Anterior–Posterior (mm)	t-Value	z-Score	KE
V2L	Somatosensory Cortex	−5.2	1.2	−6.8	4.21	3.52	229

**Table 2 brainsci-13-01705-t002:** Brain regions displaying significant decreases (*p* < 0.001, K > 50) in BGluM after chronic exercise and chronic cocaine exposure. Cluster location is indicated (medial–lateral, anterior–posterior, and dorsal–ventral). KE represents the cluster size or number of voxels in the cluster. Each box under “Brain Region(s)” indicates a separate cluster.

Brain Region	General Cluster Location	Medial–Lateral (mm)	Dorsal–Ventral (mm)	Anterior–Posterior (mm)	t-Value	z-Score	KE
Crus1, PFL	Cerebellum	−5.0	6.0	−11.4	5.92	4.45	2496
PFI, Crus1	Cerebellum	4.0	5.2	−11.0	6.18	4.57	2175
8cb, PM, Cop	Cerebellum	−0.8	4.4	−12.4	5.00	3.98	2103
St, sm, stmpm	Stria terminalis, thalamus	−1.8	5.8	−0.8	4.92	3.94	238
CPu	midbrain	−4.4	4.6	−2.4	4.68	3.80	193
VL	Thalamus	1.8	5.6	−2.4	4.44	3.66	174
S1HL	Somatosensory cortex	−2.2	1.8	−0.4	4.06	3.42	58

## Data Availability

The data presented in this study are available in article.

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
