# Peer review of "Exercise Modifies the Brain Metabolic Response to Chronic Cocaine Exposure Inhibiting the Stria Terminalis"

_brainsci, 2023, doi:10.3390/brainsci13121705_

Round 1

Reviewer 1 Report

Comments and Suggestions for Authors

The authors aimed to explore the relationship between aerobic exercise and metabolic activity in response to cocaine exposure. Their findings reveal that aerobic exercise triggers specific metabolic changes in regions associated with emotion, behavior and reward during cocaine exposure.

the dose of cocaine used (25 mg/kg) is equivalent to 1.7g daily dose for a 70kg human. This is a large daily dose with no acclimation to cocaine. Human use a wide range of doses but on average start at ~ 30-60 mg. No rationale is provided here in order to support the idea of the manuscript. Also, a low dose of cocaine (5-10 mg/kg) is needed.

the study centers on the impact of aerobic exercise history on cocaine exposure. The rats engaged in a six-week running regimen. For those exercised rats, the investigation explores potential correlations between exercise levels (possible running distance)  and the extent of metabolic changes. Additionally, are there any study examining whether the metabolic changes in those brain in response to exercise?  

The PET scans occurred 2 weeks after the cocaine exposure. Please describe the reason why?

The authors exclusively used female rats in the study, what’s the rationale for it? however, as the authors describe in the introduction (line 87-90), the effect on drug use was not observed in female students.

Author Response

the dose of cocaine used (25 mg/kg) is equivalent to 1.7g daily dose for a 70kg human. This is a large daily dose with no acclimation to cocaine. Human use a wide range of doses but on average start at ~ 30-60 mg. No rationale is provided here in order to support the idea of the manuscript. Also, a low dose of cocaine (5-10 mg/kg) is needed.

The metabolism of a rat is much greater than that of a human being due to the smaller body weight [1]. The dosing was adjusted for this difference. All methods in this experiment including dosing is in accordance with established protocol and literature.

the study centers on the impact of aerobic exercise history on cocaine exposure. The rats engaged in a six-week running regimen. For those exercised rats, the investigation explores potential correlations between exercise levels (possible running distance)  and the extent of metabolic changes. Additionally, are there any study examining whether the metabolic changes in those brain in response to exercise? We have published a paper examining the brain metabolic effects of this exact exercise protocol alone, and an additional paper looking at the effects of a single dose of cocaine. Please see Hanna et al., 2022a and 2022b. [52, 53].   

The PET scans occurred 2 weeks after the cocaine exposure. Please describe the reason why? This is standard protocol as per our previously published PET papers. Additionally, rats need at least 1 week to habituate to the new environment after they are transferred to the microPET imaging location.

The authors exclusively used female rats in the study, what’s the rationale for it? however, as the authors describe in the introduction (line 87-90), the effect on drug use was not observed in female students. Females are underrepresented in cocaine studies, and this is the third part of a larger study observing the effects of varying doses of cocaine on brain metabolism in female rats. As the reviewer stated, brain responses to drugs are sex specific and this details the effects of cocaine on females only.

Reviewer 2 Report

Comments and Suggestions for Authors

The current manuscript is an animal experiment studies the effect of exercises on brain glucose metabolic activity in response to chronic cocaine exposure rats. It finds a set of stimulatory and inhibitory effects of exercises in different brain structures. The manuscript is very interesting and well-written. Please find the following comments

1- The introduction is very long. It needs to be revised in a shorter form. Please highlight the limitations in the current evidence and what the current study will add.

2- In methods, please mention the ethical approval or the IRB.

3- Discussion

A- Please interpret the result of the activation in the secondary visual area

B- How the results can implicit for clinical field or addiction rehabilitation

C- There was no research recommendations

4- The conclusion should be short, informative, and answers the research question. Revise it, and there is no need to write about previous studies in the conclusion

Author Response

The current manuscript is an animal experiment studies the effect of exercises on brain glucose metabolic activity in response to chronic cocaine exposure rats. It finds a set of stimulatory and inhibitory effects of exercises in different brain structures. The manuscript is very interesting and well-written. Please find the following comments

1- The introduction is very long. It needs to be revised in a shorter form. Please highlight the limitations in the current evidence and what the current study will add. This introduction was expanded upon to increase the word count as per the journal’s request. As it stands, we are still below the word count. We do not wish to cut the introduction any further. Limitations on exercise as psychiatric treatment has been moved from the conclusions as per the reviewers request and added to the introduction (Line 105-111)

2- In methods, please mention the ethical approval or the IRB. This is included in the methods section already. See 1. Materials and methods: Animals (Line 176-179)

3- Discussion

A- Please interpret the result of the activation in the secondary visual area. This has been expanded upon in the first paragraph of the discussion. Please see lines 229-234.

B- How the results can implicit for clinical field or addiction rehabilitation. This research provides targets for further mechanistic brain investigation so see if artificial inhibition of these areas result in a decrease in drug preference. This interpretation has been added to the conclusion. (line 309).

C- There was no research recommendations. This was already included. Please see last sentences in conclusion, lines 312-314.

4- The conclusion should be short, informative, and answers the research question. Revise it, and there is no need to write about previous studies in the conclusion. This has been revised.

Round 2

Reviewer 1 Report

Comments and Suggestions for Authors

The authors state in line 69-70 that the “DRD2 receptor availability is negatively associated with cravings and reward seeking [28].” However, the cited paper did not fully support this idea. To accurately reflect the relevant literature, the appropriate references should be acknowledged (PMID: 17332411, 11802171, 16829955). Furthermore, a recent study (PMID: 34506723) shows that DR2D receptor sensitivity is also associated reward seeking behaviors. In light of this, it is more precise to state that DRD2 receptor availability (PMID: 17332411, 11802171, 16829955) and sensitivity(PMID: 34506723) is associated with rewarding seeking behavior.

Author Response

This statement has been rephrased in the manuscript and the suggested references have been added. Thank you!

Reviewer 2 Report

Comments and Suggestions for Authors

The authors address all required comments. 

Author Response

Thank you for reviewing our manuscript!